# Inside the Biology of the β3-Adrenoceptor

**DOI:** 10.3390/biom14020159

**Published:** 2024-01-29

**Authors:** Amada Pasha, Annalisa Tondo, Claudio Favre, Maura Calvani

**Affiliations:** 1Department of Pediatric Hematology–Oncology, Meyer Children’s Hospital IRCCS, 50139 Florence, Italy; amada.pasha@unifi.it (A.P.); annalisa.tondo@meyer.it (A.T.); claudio.favre@meyer.it (C.F.); 2Department of Experimental and Clinical Biomedical Sciences, University of Florence, 50139 Florence, Italy

**Keywords:** β3-adrenoceptor, cancer, hypoxia, retinopathy, prematurity, Ewing sarcoma, neuroblastoma

## Abstract

Since the first discovery in 1989, the β3-adrenoceptor (β3-AR) has gained great attention because it showed the ability to regulate many physiologic and metabolic activities, such as thermogenesis and lipolysis in brown and white adipose tissue, respectively (BAT, WAT), negative inotropic effects in cardiomyocytes, and relaxation of the blood vessels and the urinary bladder. The β3-AR has been suggested as a potential target for cancer treatment, both in adult and pediatric tumors, since under hypoxia its upregulation in the tumor microenvironment (TME) regulates stromal cell differentiation, tumor growth and metastases, signifying that its agonism/antagonism could be useful for clinical benefits. Promising results in cancer research have proposed the β3-AR being targeted for the treatment of many conditions, with some drugs, at present, undergoing phase II and III clinical trials. In this review, we report the scientific journey followed by the research from the β3-Ars’ discovery, with focus on the β3-Ars’ role in cancer initiation and progression that elects it an intriguing target for novel antineoplastic approaches. The overview highlights the great potential of the β3-AR, both in physiologic and pathologic conditions, with the intention to display the possible benefits of β3-AR modulation in cancer reality.

## 1. Introduction

After the first classification of β-adrenoceptors (β-ARs) [1] and the identification of the third subtype, β3-AR, in 1989 [2], the attention of research studies focused on the involvement of β3-ARs in physiologic and pathologic conditions, with particular attention focused on cardiovascular disease, diabetes, obesity and cancer.

In the last two decades, it has been largely demonstrated that stress-related catecholamines (CA), such as noradrenaline (NA) and adrenaline (A), are involved in tumor neurogenesis; in particular, CA could promote the acceleration of cancer progression, which reduces the overall survival of patients [3,4]. Several studies have reported that stress is connected to tumorigenesis and cancer progression. Under stressful conditions, the secreted CA hormones, A and NA, bind to β-ARs and activate downstream signaling pathways that induce tumor cell growth and metastasis [5]. β-ARs signaling, through molecular and cellular processes, affects tumor growth, progression and metastasis [6,7]. In addition, the CA binds to β-ARs, localized on the immune cell surface in the TME, and regulates the innate and adaptive immunity [8].

Except for pathologic conditions, it has been largely demonstrated that β3-ARs have important physiological effects, such as lipolysis and thermogenesis regulation in adipocytes through the β-oxidation of fatty acids [9]. β3-ARs could induce both the smooth and detrusor muscle relaxation in the urinary bladder [10], as well as vasodilation and relaxation to cardiac contractility [11]. β3-ARs are highly expressed both in the pregnant myometrium [12] and during early embryo development [13,14]; conversely, in healthy adult human tissues, β3-ARs exhibit a restricted expression (brain, retina, blood vessels, myocardium, myometrium, adipose tissue, gallbladder, and urinary bladder). It has been proved that β3-ARs are over-expressed in the TME and that they are involved in angiogenesis, cancer progression and tumor stromal cell reactivity [15,16]; therefore, the targeting of β-ARs has been suggested as a possible therapeutic approach for cancer treatment.

## 2. β-Adrenoceptors

The last two decades have witnessed increased interest in β-Ars’ role in several physio-pathological conditions. Experimental modes have showed that β-AR blockers may be efficient in reducing cancer growth and progression.

The β-AR system consists of CA, A secreted by the adrenal medulla, and NA secreted by local sympathetic nerve fibers, and their receptors classified, respectively, in the alpha (α) and beta (β) adrenoceptors that are widely expressed in mammalian tissues with important roles in physiological processes [17]. A and NA are classic neurotransmitters mediating fight-or-flight stress responses that bind to β-ARs with different affinities [18]. β-ARs are members of the 7-transmembrane (TM) receptor family, which belongs to the superfamily of G-protein-coupled receptors (GPCRs) with a pocket for the binding site of the ligands [19]. GPCRs show three intracellular and extracellular loops, an extracellular N-terminal domain, and a cytosolic C-terminal tail with phosphorylation sites for GPCR kinases (Figure 1). 

β-AR phosphorylation is involved in the rapid desensitization and downregulation of the receptor [20]. The ligand–receptor binding induces a conformational change in the C-terminal tail of the receptor that couples with a G-protein with an exchange of guanosine diphosphate (GDP) with guanosine triphosphate (GTP) and separation of the G-proteins into active G*βγ* and Gα subunits. The downstream effects of GPCR activation depend on the Gα subunit type (G*α*.S, G*α*.I, G*α*.q, and G*α*.12) to which the receptor is coupled [21]. The ligand stimulation of β-adrenoceptors activates the Gs subunit where the active α subunit binds to the adenylyl cyclase (AC) that converts ATP into cyclic 3′-5′-adenosine monophosphate (cAMP) [22], which in turn activates the cAMP-dependent protein kinase A (PKA), with a phosphorylation of serine/threonine residues of β-AR kinase (β-ARK). β-ARK kinase phosphorylates the serine/threonine residues of the adrenoceptor, providing the binding site for β-arrestin to the adrenoceptor. The binding of β-arrestin to the adrenoceptor prevents further coupling to G proteins and promotes receptor internalization. In the end, GTP is hydrolyzed into GDP via the GTPase activity of the α subunit that reverts the receptor to its original state and allows it again to interact with the βγ subunits (Figure 2) [23]. β-ARs are classified into three subtypes: β1-ARs, β2-ARs, and β3-ARs. β1-Ars, which are localized mainly in the striatum cardiac muscle, airway muscles and juxtaglomerular apparatus, show a similar affinity for both NA and A binding. β1-ARs are excitatory receptors whose activation leads to positive cardiac ionotropic and chronotropic effects, while in the juxtaglomerular cells of kidney, they promote renin secretion, and in adipocytes, they stimulate lipolysis [24]. 

β2-ARs, which display a higher affinity for A, are localized in the liver, skeletal muscle and in gastrointestinal and bronchial smooth muscle cells. β2-Ars’ activation induces bronchodilation and muscle relaxation, redirecting the blood flow and mobilizing energy accumulation [25]. In immune and non-immune cells, β2-ARs are involved in immunoregulation and immune response [26,27], but they are also expressed in blood vessels and coronary arteries, where they promote the increased irrigation of the organs. The β2-AR is generally involved in carcinogenic processes. Polymorphisms of the gene encoding β2-AR, *ADRB2*, have been associated with several diseases, such as asthma, obesity and cancer [28]. 

The last subtype identified, β3-ARs, are excitatory receptors, localized in adipose tissue (both BAT and WAT), vascular endothelium, and the small intestine [29], and they differ from the β1- and β2-AR subtypes with respect to the sequence and pharmacological profile. Compared to β1- and β2-ARs, β3-ARs show a limited expression pattern in humans [30]. The β-ARs signaling pathways, including the β3-AR subtype, imply AC stimulation, which induces cAMP production [24]. The gene encoding β3-AR, *ADRB3*, has been identified in several species, such as mouse [31], rat [32], sheep, bovine, goat [33] and dog [34]. In humans, *ADRB3* is found on chromosome 8, and it shares 51% and 46% of sequence homology with β1- and β2-AR amino-acid sequences, respectively [35]. 

The ADRB3 gene is made of a single intron that corresponds to a polypeptidic chain of 408 amino acid residues, and together with β1- and β2-ARs, it belongs to the GPCR family with 7-TM domains that contain a glycosylated extracellular N-terminal region and an intracellular C-terminus [36]. 

β3-AR has three intra- and extracellular loops, but it lacks the phosphorylation sites for both PKA and β-ARK kinases. The C-terminus is the region with the lowest homology and is characterized by the lack of phosphorylation sites, which are involved in the agonist-dependent desensitization of the β1- and β2-ARs isoforms [37]. The lack of phosphorylation sites, therefore, makes β3-ARs resistant to agonist-induced desensitization [36]. Indeed, among others, this characteristic makes β3-AR an interesting therapeutic target that possibly could be approved for chronic treatments [38].

The β3-AR subtype has been largely correlated with metabolic regulation, both in physiological and pathological conditions [39]. β3-AR inhibits the contractile activity of the ileum and colon [40], and it controls neuronal broncho-motor with airway smooth muscle relaxation [41]; moreover, it induces peripheral vasodilatation, reduces contractile force in human ventricular muscle and in human atrial myocytes, it stimulates L-type calcium current. In the last few decades, β3-AR has been proposed as a potential therapeutic target in pathologies such as cachexia, diabetes, obesity, and cardiac disease [42]. In addition, to date, despite the great number of ligands that have been tested, their assignment to either one or the other category, among the receptors’ agonists or antagonists, is still under debate [38,43].

## 3. β3-AR Physiology 

Since their discovery in 1989, the β3-ARs’ physiology and function have been rapidly studied. The involvement of β3-ARs in cancer initiation/progression suggested that they could be an interesting target for novel anti-tumoral and therapeutic approaches. Furthermore, studies have proved the pleiotropic roles of β3-ARs, which appear to be involved in different type of cancers, where they promote similar benefits. 

However, given the late discovery, β3-ARs’ role in humans is still controversial. Advances over the last two decades have proposed the targeting of β3-ARs for the treatment of several conditions, with some drugs undergoing phase II and III clinical trials and others already available for the treatments [44]. 

### 3.1. β3-AR in Adipose Tissue

BAT is the main thermogenic organ involved in the cold response, which improves the energy expense (non-shaking thermogenesis) or nutritional excess (diet-induced thermogenesis) in mammals [45]. The mRNA transcript of β3-AR was identified in infant peri-renal BAT and in several deposits of WAT in adults [31]. Pharmacological activation of β3-ARs in BAT and WAT in mice models inactivated both thermogenesis and de novo lipolysis [46,47].

Thermogenesis in human BAT has physiological relevance for the inverse correlation between active BAT and obesity [48,49]. β3-AR expression, which is important for the energy balance, has been observed in some brain areas, such as the hypothalamus and brain [50]. In addition, the injection into the central system of the β3-AR agonist activated some areas within the hypothalamus [51] that are recognized for their role in the regulation of appetite and body weight [52]. The β3-ARs’ regulation of appetite was described in the study of Tsujii et al., where the acute central administration of the β3-AR agonist decreased food intake and body weight in obese rats [53], showing thus a possible role of β3-ARs in food consumption and body weight regulation. In previous studies, after central and peripheral drug administration, β3-AR stimulation reduced the appetite [54,55].

In the BAT of rodents, the thermogenic activity was activated through the β3-ARs’ stimulation, which are abundant on mature brown adipocytes [54,56]. The adipocytes, through the CA effects on β-ARs, control the energy storage while eating and the energy expenses during fasting through the increase in lipolysis and fatty acid oxidation (FFA). 

In BAT, β3-ARs, through Gsα signaling [57], promoted lipolysis and thermogenesis via the mitochondrial uncoupling protein 1 (UCP1) stimulation that uncoupled the mitochondrial oxidative phosphorylation. This process produced a proton conductance pathway across the inner membrane, with an increase in energy production [58,59]. In mice models with the Gαs subunit deleted [60] (adipo-Gαs-KO knockout), it was observed the lipolysis of adipocytes through β3-AR stimulation of the cAMP-PKA pathway, which improved perilipin phosphorylation and triglyceride hydrolysis in response to β3-AR agonism and reduced the FFAs levels [61]. In adipose thermogenesis, the increase in β3-AR expression revealed the important role of cAMP signaling, where the thermogenic genes, such as *Ucp1*, *Cidea*, *PPARγc1α*, *PPARγ2*, and *Pdk4*, were suppressed in the WAT and BAT of Adipo-Gsα-KO-mice [61,62]. Moreover, β3-AR signaling controls the carbohydrate and lipid metabolism [63].

Under low temperatures, the CA release and β3-AR activation transmitted the signal to BAT, which increased the UCP1 expression and activity [64]. In beige adipocytes, in the cold response, the PKA stimulated β3-ARs, which directly phosphorylated mTOR and its binding partner, a regulatory-associated protein of mTOR (RAPTOR), which increased the mTORC1 activity, the key regulator of autophagy [64].

In addition, β3-ARs’ activation increased the transcription/expression of the inducible nitric oxide synthase (iNOS), which induced lipolysis that induced the nitric oxide (NO) levels in a PKA-dependent way [65].

Studies have proved the expression of β3-AR mRNA levels both in human BAT and derived adipocytes [66,67], where the selective agonist, mirabegron, increased the BAT metabolic activity and whole-body energy consumption other than the plasma FFA and BAT glucose uptake [66,68]. Moreover, ADRB3 gene mutations were associated with an increased risk of obesity and diabetes, insulin resistance and nonalcoholic fatty liver disease in the obese condition [69,70]. 

The correlation between β3-ARs and UCP1 has been largely studied in BAT [46]. For these effects, β3-ARs agonists are considered good candidates for obesity treatment [71]. Cero et al. described the importance of β3-ARs in human brown/beige adipocytes to maintain the multiple components of the lipolytic and thermogenic cellular machinery [72], since the β3-AR silencing compromised the genes essential for FFA metabolism mitochondrial mass and thermogenesis. The β3-AR-KO showed an impaired cold-induced thermogenesis in mice that reduced the white adipocyte beiging [73]. However, recent studies reported that β3-AR-KO increased the high fat diet, inducing more severe obesity, with WAT hypertrophy and inflammation, compared to wild-type mice [74].

Furthermore, in adipose tissue, some studies revealed that β3-AR stimulation had effects similar to PPARγ transcription factor stimulation [75], which exerts antioxidant properties [76,77], although no studies have demonstrated the β3-AR activation of PPARγ [78,79]. Interestingly, it has been underlined that a genetic relationship exists between β3-ARs and PPARγ genes, since they have evolved in concomitance with primates to adapt to cold temperatures [78]. Prdm16 is an important transcription factor that regulates the process that turns WAT into beige tissue [80], and it has been linked with mitochondrial defects and changes in transcriptional pattern, since Prdm16 absence implied a switch from fatty acid oxidation to glucose consumption [81]. In adipocytes, the β3-AR signaling upregulated the expression of Prdm16 [82], while inversely, the Prdm16/ebpβ complex improved the transcription of β3-ARs through the transactivation of the β3-AR promoter [83].

It is noteworthy that, at present, any of the agonists have undergone beyond phase II of clinical trials for the treatment of obesity and metabolism in adults, maybe for the lack of oral bioavailability or for the side effects caused by β3-ARs agonists. 

### 3.2. β3-AR in the Myocardium

In the myocardium, β-AR pathways are involved in the cardiac response to acute/chronic stress. In adipose tissue, besides regular lipolysis and thermogenesis, it has been described the presence of the β3-AR transcript [84] and protein [85,86] in cardiac tissue, both in the atria and ventricles. In 1996, it was reported the expression of β3-AR transcripts in biopsies of human hearts [87].

In cardiac muscle, β2- and β3-ARs are coupled to Gi, whose triggering caused an increase in NO through endothelial nitric oxide synthase (eNOS) activity, which is expressed both in endothelial cells and in the ventricular cardiomyocytes [88], which increased the NO production. Contrary to β1/β2-ARs, which are downregulated and desensitized under CA stimulation, the expression of β3-ARs was upregulated in the myocardium of patients with heart failure (HF) [89] and in HF of animal models [11,90]. Evidence has shown the beneficial effects of β3-ARs in the endothelium and myocardium, with effects in the cardiovascular system, including cardio-protection [91]. 

The link of β3-AR to intracellular effectors is different among tissues and cell types. β3-ARs’ expression differs in the vasculature and vessels sizes of different species. β3-ARs’ activation produced vasorelaxation, but the signaling was different in various cell types. In canine pulmonary arteries, the β3-ARs vasodilating effects were coupled with increased cAMP levels [92]. In endothelial cells, the β3-AR activation produced the relaxation of the endothelium in aortic and resistance vessels, both through NOS-dependent and -independent mechanisms [37]. 

The agonism of β3-AR induced negative inotropic effects in the myocardium and vasodilation of blood vessels [37,93]. Other than the vasodilatory effect, the β3-ARs’ activation produced higher levels of NO that had a proangiogenic role in a limb ischemia model induced by diabetes [94]. In human atrial myocytes, β3-ARs stimulated an L-type calcium current that caused the calcium channels’ phosphorylation and I_Ca_ to increase [95]. 

After an ischemic injury, to preserve cardiac output, it has been observed an increased sympathetic activity and increased release of CA that stimulated the β-AR-mediated inotropic capacity.

In human myocardial tissue, the lack of β3-AR sites for GPCR kinases, which causes the absence of desensitization or downregulation signaling, revealed that in the failing myocardium, the β3-AR levels did not change but were increased, maintaining, thus, their intracellular signaling [96]. In the heart, GPCRs represent the main modulators of both function and morphology, with β-ARs being “the heads of the line”, which are considered the most important molecular targets in the cardiovascular system [29,97].

The β3-ARs’ role in the heart was under debate for a long time. Some studies have suggested that for its cardio-depressant effect, the prolonged activation of β3-ARs in HF could reduce the cardiac function, since the improvement of β3-ARs could represent either a protective mechanism against the harmful effects of chronic β-AR stimulation or it could be a negative mechanism that may induce additional deterioration of the HF [98]. The main mechanism responsible for the β3-AR-induced cardio-protective effect was shown to be the NOS activation and production. In another study of animal models, with small (mice) and large (pigs) dimensions for ischemia/reperfusion (I/R) injury, the administration of the selective β3-AR agonist affected positively the infarct size (acutely) and left ventricular function (chronically) [99]. Moreover, the β3-AR/NO signaling showed a reduced permeability in the mitochondrial transition pores, resulting in the protection of cardiac cells against cell death.

In addition, it has been shown the role of exercise in cardio-protection [100], and it has been proved that β3-ARs have a mediator effect, particularly in a setting of cardiac I/R injury [101]. During exercise, the Gαs subunit-mediated protection induced by β3-ARs that caused the PKA/Akt/eNOS activation, suggesting that, in some conditions, β3-ARs were coupled with both Gαs and Gαi in cardiomyocytes [102]. Other preclinical studies have shown that β3-ARs could activate different signaling pathways for heart protection. In response to chronic β1-AR blockage, myocardial β3-AR coupling, signaling and expression were improved [29,90].

In a dog model of rapid HF, correlated with increased β3-AR expression, the acute β3-AR stimulation further decreased the contractility and the transient Ca2+ of isolated ventricular myocytes from FH [11]. An analogous negative inotropic effect of β3-AR stimulation was reproduced in vivo in dogs with HF upon infusion of the β3-AR agonist [103]. The upregulation of β3-ARs was also reported in human diabetic hearts [104], where it could affect the altered cardiac inotropic response in diabetic cardiomyopathy [105]. The cardioprotective potential of β3-AR against HF was determined first from the KO-mice phenotype that lacks the receptor [106]. In a different model of myocardial infarction, the systemic infusion of β3-AR agonist maintained the cardiac contractile function, with reduced fibrosis and apoptosis [107]. 

The expression of the β3-AR mRNA levels was significantly increased in a rat model with an isoprenaline-induced cardiac hypertrophy [108]. In another study, in a rabbit model, the preservation of the Na^+^/K^+^ pump function, upon β3-AR stimulation, promoted the reduction of congestive cardiac remodeling, which was induced by coronary ligation [109], a phenomenon observed in a diabetic heart model, where the stimulation of β3-AR prevented the Na^+^/K^+^ pump inactivation [110]. 

The cardio-depressant effect of β3-AR was showed in another study by Bosch et al. [111], where in ventricular myocytes of guinea pig, the selective β3-AR agonist or isoprenaline, in the presence of β1 and β2-AR blockers, inhibited IKs (while isoprenaline alone stimulates this current), modifying the length of potential action. In this species, this result may partly explain the inhibitory effect of β3-ARs on heart function [112]. 

In some species, the pulmonary arterial vasculature express β3-AR,s where they contributed to reduced pulmonary vasoconstriction and reduced pre-capillary pulmonary hypertension [92,113].

In the last few years, evidence has supported the existence of a fourth controversial β-AR subtype in the human atrium, the β4-AR, which has been shown to be coupled to Gs-AC signaling [87,114,115]. The β4-AR presence has been supported by results obtained from the CGP-12177, a β3-AR agonist developed initially as a β1/β2-AR antagonist but shown later to be a partial β3-AR agonist [116]. In β3-AR-KO-mice, the β4-AR subtype was the crucial mediator of the cardiac and thermogenic responses induced by CGP-12177 [117,118]. Therefore, these experimental proofs indirectly suggested that β4-AR expression was probably limited to adipose and heart tissues [119]. In this regard, to date, research studies are focusing on the study of the specific role of this new receptor subtype.

### 3.3. β3-AR in Urinary Bladder

Besides the myocardium, in the 1970s, some studies suggested the presence of a peculiar β-AR in human detrusor [120,121], which, through adrenoceptor signaling and sympathetic nerve activity, promoted urine storage by relaxing the detrusor muscle. The most important adrenoceptor involved in bladder relaxation, in the storage phase of the micturition cycle, was showed to be β3-AR [122,123]. In the urinary bladder, the sympathetic-mediated relaxation in humans, monkeys, and dogs was dependent on β3-AR activation [124,125,126]. These studies were confirmed by the proof that muscle relaxation in pre-contracted samples could not be induced only with β3-AR agonists rather than β1- and β2-AR agonist stimulation. Through β3-ARs, which are expressed in the smooth muscle cells of the urinary bladder, the sympathetic system could mediate the detrusor muscle relaxation, improving the filling capacity of the bladder via cAMP signaling [123,127]. In particular, β3-ARs are found in the lower urinary tract [43] and urethra [128], where they regulate urothelium and prostate function [129,130], where it seemed to be the predominant subtype [131]. This promoted the improvement of specific β3-AR agonists that are currently used in clinics for the overactive bladder treatment [132], as they were well tolerated and really improved bladder symptoms [133].

Except for the human urothelium, β3-AR expression has been observed also in different species, such as rat [134] and pig [135]. About β3-AR signaling pathway in the bladder have been suggested different, and not always coherent, hypotheses. 

Commonly, β-AR signaling depends on the cAMP/PKA pathway [92,136], where cAMP is a key player in smooth muscle relaxation. However, its involvement in this process is still a matter of debate.

D’Agostino et al. proved that β3-AR activation, after electrical field stimulation in human detrusor bladder strips, clearly inhibited contraction and acetylcholine (Ach) release from cholinergic nerves in a concentration-dependent manner [137]. In support of the fact that β3-AR modulated the ACh release, a recent study showed, via immunohistochemistry, that β3-AR was expressed in the detrusor cholinergic fibers near sympathetic bundles that innervate the human bladder. This conformation was physiologically reasonable, as adrenergic fibers could release NA that activated β3-ARs [138]. In addition, the systemic activation of β3-ARs reduced the activity of the mechano-sensitive Aδ afferent fiber and the hyperactivity prostaglandin E2-induced C-fiber [139]. 

At present, among the β3-AR agonists, mirabegron has shown to be the most promising treatment in over 30 years for the oral pharmacotherapy in patients affected by overactive bladder syndrome. This adrenoceptor’s agonist has been widely studied, with several phase II and III trials showing a good ratio between efficacy and tolerability [140]. For the promising properties, mirabegron has been approved also for children’s treatment with neurogenic detrusor overactivity [141]. There is a new potent and selective β3-AR agonist, vibegron, that has made it to phase III of clinical trials in adults. It has recently passed the latter stages [142], and it has been authorized in Japan for overactive urinary bladder (OAB) treatment [143]. 

Recently, in Japan, mirabegron has been authorized for OAB treatment, and it has shown in clinical studies fewer cardiovascular adverse effects [144]. Kanie et al. have proposed another promising compound for OAB treatment, the selective human agonist TRK-380, which functionally induced relaxation in isolated detrusor strips of several mammalian species [145]. In human bladder strips, TRK-380 showed a relaxing effect on both the resting and contractile responses, and the relaxation was mediated principally by β3-AR activation. Moreover, TRK-380 showed minor or no activity for human β1- and β2-ARs and, as with mirabegron, it might cause less cardiovascular effects. However, to verify the TRK-380 effect on cardiovascular outcomes, additional in vivo experiments in vascular tissue are needed. Previous β3-AR agonists, such as BRL-37344 and CL316,243, were withdrawn since they were efficient in rodents but not in humans for the species differences in the responses [134,146]. 

Hence, it is very important to precisely assess the selectivity and effectiveness of compounds for human β3-ARs in the preclinical stage. Unfortunately, at present, there is a lack of literature data that describe the use of β3-AR agonists in pediatric populations.

### 3.4. β3-AR in the Myometrium and Pregnancy

In pregnancy, the role of β-AR signaling was extensively reported [147]. In the fetal development under hypoxic conditions, CA modulates fetal circulation through the reduction of the fetal heart rate and by preserving heart and brain glucose homeostasis [148]. The pregnant myometrium and early embryo development [13,149,150] shared similarities between the environments surrounding the tumor and embryo. Under hypoxic intra-uterine life, they both highly expressed β3-AR in the developing embryo, while it decreased after birth [151] when newborns are suddenly exposed to oxygenated ambient atmosphere. 

Hypoxia, which plays a great role in fetus development and cancer progression, is involved in angiogenesis, apoptosis, cell migration, invasion, and metastasis processes [152]. Early human placental tissue matured under a hypoxic environment necessary to induce specific placental metabolic activities [153]. The premature contact of immature organs with oxygenated ambientes, which have a moderate hyperoxic environment compared to physiologic intra-uterine hypoxia, could control their growing process by reducing β3-AR expression and impairing its role in organ maturation [154]. 

The contribution of β3-ARs in promoting embryo properties and functional roles in germ cells, in preimplantation embryos and in the first week of embryogenesis, suggested that the original function of β3-ARs could have been involved in the promotion of proliferation, vascularization, metabolic adaptation and immune tolerance [155]. Since β3-AR was expressed under hypoxia, it has been assumed a relationship between the hypoxic upregulation of β3-ARs, which promoted immune tolerance [156], and the initial mechanisms that were predicted to guarantee fetal tolerance. In the female reproductive system, such as the placenta, ovaries, and fallopian tubes, there was reported the expression of β3-AR mRNA and protein levels [157], but also in mammalian oocytes [158], where they induced motility [159]. In Chinese hamster ovary/K1 cells, it has been reported the β3-ARs expression [29]. Activation of β3-AR in the myometrium relaxed contractions [12], and its expression increased during gestation [160]. In myometrial microvascular endothelial cells, mirabegron stimulated the NO production and the uterine tissue relaxation, in vitro*,* was partially blocked by the addition of the eNOS synthase blocker [161].

The presence and function of β3-ARs in the human near-term myometrium has been described [147]. β3-AR stimulation induced significant relaxation of myometrial contractions via a cAMP/PKA mechanism [148]. However, the β3-ARs’ role in embryonic development and fetal life remains unclear. The inverse of β2-ARs, β3-ARs were less inclined to sustained agonist-induced desensitization in the near-term myometrium [12]. In the same study by Rouget et al., the evidence of β3-ARs’ presence in the myometrium and their upregulation in pregnancy was strengthened by showing that they were the main subtype both in the human non-pregnant and pregnant myometrium (10- and 20-fold more abundant, respectively). Moreover, functional experiments supported that the selective β3-AR agonist, SR 59119A, strongly inhibited spontaneous contractions in the pregnant than the non-pregnant myometrium. 

In humans [157] or rats [162], pregnancy might influence β-AR expression. It is well documented the influence of sexual hormones on β2-ARs’ expression in the myometrium. In specific, progesterone, through the control of the transcription rate of β2-AR gene in late-pregnant rats, may increase the density of the receptors’ expression [163,164]. In guinea pig, estradiol markedly upregulated the β2-AR density, while progesterone influenced β2-AR to a minor intensity [165]. In comparison, little is known about the steroid hormones and their modulation of β3-ARs. In human pregnancy, there is a progressive improvement in the myometrial 17β-estradiol to progesterone ratio, and interestingly, the β3-AR expression increased in the near term in the myometrium in response to an agonist compared to the non-pregnant myometrium [166]. In brown adipocytes, estradiol and progesterone modified β3-AR affinity and density [167]. These results led to the hypothesis that the hormonal changes in pregnancy could probably explain the variation of β2- and β3-AR expression, their functionality and affinity compared with non-pregnant myometrium. These data suggested a potential role of β3-ARs in the regulation of preterm labor. β3-AR induced motility [159] in pre-implantation embryos [158,168], in the first stages of embryogenesis [13], in embryo tissues and in the placenta [149,150] of human and animal germ cells. Also, in the human pregnant myometrium, β3-AR was upregulated and it inhibited spontaneous contractions, representing the predominant subtype over β2-AR [169]. In addition, the β3-AR agonist induced the adequate vasodilation in the human umbilical artery ring, implying that β3-ARs may be involved in the regulation of fetoplacental circulation [170]. These results show the role of β3-ARs in promoting fecundation, embryo implantation and growth. 

Lately, studies have revealed important similarities between cancer growth and embryo development [152], since they share hypoxic conditions and an acidotic environment [153,171]. The evidence that, together, cancer and embryo expressed β3-ARs upregulated under hypoxia guided researchers to the hypothesis that β3-ARs over-expression might represent one of the mechanisms by which in origin embryo, and then cancer, adapted their growth in an adverse environment [172]. The stimulation of β3-ARs inhibited the cytokine production (such as TNFα, IL-6, and IL-8), which prevented myometrial cell apoptosis [173]. On the other side, β3-AR stimulation induced human myometrial cell proliferation that preserved cells in an undifferentiated non-contractile phenotype [174]. Another interesting pathway is the antioxidant role of β3-ARs in the macrophages of the myometrium [175]. β3-AR has been also linked to another pathway, such as BKCa, the potassium channel activated by calcium-conductance [176], which is the most important potassium channel in the myometrium, where β3-AR played a key role in the potential regulation of the cell membrane and in myometrial quiescence [177] (Figure 3). 

In conclusion, still, the mechanisms that underly the β3-AR-mediated relaxation in the myometrium are not completely clear.

## 4. β3-AR in Pathology

Cancer is one of the principal causes of death in the world. At present, researchers focus the attention on the role of sympathetic nerves in cancer development (through CA and adrenoceptors) [138]. It has been well proved the correlation between β-ARs and cancer progression, including inflammation, angiogenesis, cell motility and trafficking, apoptosis and cellular immune response. Particularly, the activation and expression of β3-ARs in stromal and tumors cells of the TME have been shown to stimulate pro-tumoral signaling pathways, which caused tumor progression in different pre-clinical models [4,15,156].

The first evidence on β-ARs’ role in tumor growth came from Schuller et al.’s studies [178], where they revealed that a non-selective β-AR agonist, isoprenaline, induced the lung adenocarcinoma cells’ proliferation, while propranolol counteracted this effect. The pro-tumoral role of β-ARs was later showed in several types of cancer, such as pancreas, breast, ovary, colorectum, esophagus, stomach, lung, prostate, melanoma, leukemia, hemangioma and angiosarcoma [179]. 

Soon enough, β3-ARs were incredibly attractive in cancer biology. Considerable amounts of β3-ARs have been detected in a range of malignant tumors, such as melanoma, vascular tumors, leukemia, colon and breast cancer [16,180,181,182]. Pharmacological evidence showed the presence of β3-ARs in human ovarian and endometrial cancers [183]. 

The literature data reported the β3-ARs’ ability to reduce tumor growth and metastases. Bruno et al. showed that β-ARs, principally through β2- and β3-ARs, supported tumor growth and cancer pain in a murine model of syngeneic osteosarcoma (OS) [184].

Under hypoxic conditions, β3-ARs were over-expressed in the TME and increased cancer growth with the recruitment of circulating stromal cell precursors to the tumor sites, together with improvement of stem cell traits [185]. In response to the tumor cell proliferation of solid tumors, the β3-ARs’ upregulation led to the deprivation of oxygen and nutrients; this resulted in the activation of transcription factors involved in the induction of tumor cells to escape from an oxygen-deprived environment [186]. Curiously, β3-AR expression has been identified in human leukemia [181], where it was involved in myeloid leukemia cell survival under hypoxic conditions to such a quantity that β3-AR could be considered a potential target to reduce chemoresistance, a phenomenon that occurs frequently in leukemic patients [187]. An important challenge for the investigation of metastatic and localized diseases includes the identification of markers that can detect recurrences and metastasis in cancer.

### 4.1. β3-AR in Melanoma

For melanoma therapy, targeting β-ARs has been proposed as a potential therapeutic approach [3,188]. Patients with melanoma who were treated with β-AR blockers have reported a considerable decrease in the risk of melanoma progression [188,189,190]. Meta-analysis and preclinical evidence have shown that among the β-AR blockers, propranolol acted mainly on β1/β2-ARs [43,191]. Originally, the attention was largely focused on β2-ARs [192], but recently, the β3-AR over-expression was found in melanoma cell lines and TME [16,193]. Recently, Dal Monte et al. have proposed a role for β3-ARs in melanoma tumor [194]. In another study, it has been sustained that β3-ARs are expressed by all the type of cancers and mostly in melanoma [195]. Moretti et al. have reported that primary melanoma cells express both β1- and β2-ARs, and that β2-ARs in particular were upregulated in metastatic melanoma, which strongly correlates with malignancy [192].

There have been difficulties with the development of clinical studies due to the poor pharmacological profile of β3-AR blockers and the limitation on the understanding of β3-AR functions. One of the reasons why this subtype of adrenoceptor has been poorly investigated is its restricted expression profile compared to β1- and β2-ARs [38]. 

The presence of β-ARs was detected on the surface of stromal and inflammatory cells, but also on the vascular cells of the TME in melanoma, where β3-ARs could help with the environmental stimuli response that increases cancer cell motility and induces stem-like traits [16]. Moreover, in the melanoma B16F10 cell line, β3-AR was expressed and significantly upregulated under hypoxia, increasing the production of vascular endothelial growth factor (VEGF) in a NO-mediated way. In a mice model of melanoma, the β3-AR blockade with intra-tumor injections of SR59230A, the β3-AR antagonist, reduced tumor growth and vasculature through melanoma cells apoptosis, showing, thus, the involvement of β3-ARs in melanoma cell survival and proliferation [15]. In human samples of melanocytic nevi, the β3-AR expression has been identified in primary melanoma, superficial spreading melanoma, cutaneous, nodular and lymph-nodal metastatic melanoma [196]. 

Melanoma showed an over-expression of β3-ARs, which exhibited a clear correlation with malignancy and advanced malignant lesions [16]. The possible pro-tumorigenic role of β3-ARs has been studied in a C57BL/6 mouse model of melanoma where syngeneic B16F10 cells were inoculated after β1- and β2-ARs gene deletion. In this study, the SR59230A or L-748337 administration reduced the tumor volume, weight and vascularization [197]. This showed that the β3-ARs’ effects on melanoma growth were sustained by the β1/β2-ARs activity expressed by the host. In addition, the block of β3-ARs could neutralize melanoma growth, with other effects on melanoma micro-environment. Interestingly, Calvani et al. have shown that β3-AR signaling had a key role in the crosstalk relating the stromal compartment and tumor cells [16]. In fact, in the A375 melanoma cell line, under a conditioned medium from activated stromal cells with M2 macrophages, myofibroblasts and hypoxic endothelial cells, there were observed higher β3-ARs levels rather than β2-ARs. These data showed that β3-ARs are involved in the angiogenic processes that induced melanoma cells to respond to environmental cell signals, such as VEGF and inflammatory cytokines. Moreover, in A375 cells in the conditioned medium from human dermal fibroblasts, there was an increase in the stem-like markers of melanoma cells, which were inhibited by SR59230A but not by β2-AR blockade, showing thus that β3-ARs were the main receptors involved in the acquisition of stem-like traits of melanoma cells.

Recently, Calvani et al. [198] confirmed the antioxidant role of β3-ARs. In detail, they showed the β3-ARs expression in the mitochondria of embryonic stem cells (ESC) and cancer stem cells (CSC), where the β3-ARs stimulation with the BRL37344 agonist induced the Warburg effect, which consisted of an accelerated aerobic glycolysis. The β3-AR–Warburg effect involved UCP2, implicated in the mitochondrial ROS (mtROS) content modulation and whose expression was inhibited by SR59230A. It was proved that SR59230A and the specific UCP2 inhibitor, Genipin, increased the mtROS content in both CSC and ESC treated with BRL37344, with a great increase in CSC. 

These results confirmed the data literature, where β3-ARs and UCP2 had a strong antioxidant role [175,199], and clearly suggested that the β3-ARs/UCP2 axis promoted mitochondrial dormancy through the inhibition of ATP production and the mtROS content that led to increased aerobic glycolysis of cells. 

Moreover, the involvement of β3-ARs in immune cells has been studied. T-lymphocytes express both β2- and β3-ARs, among them β3-ARs were mainly upregulated in response to stress [200]. In melanoma, β3-ARs’ role in regulating the immune tolerance was evaluated through the analysis of its antagonism on cytotoxic and suppressive immune cell sub-populations [156]. 

Initially, β2-AR expression has been demonstrated in immune cell sub-populations such as NK [201], CD8 [202], and Treg [203] cells. Later, β3-AR antagonism reduced melanoma growth in vivo by increasing the NK and CD8 cell numbers as well as their cytotoxicity in the TME through the regulation of Treg and MDSC sub-populations [156]. These findings were coherent with the effect of β3-AR agonism that inhibited the pro-inflammatory (M1) activity of macrophages [175] and with previous studies that showed the phenotypic plasticity of macrophages and neutrophils in the TME [204].

In conclusion, the β3-ARs’ involvement in the pathophysiology of melanoma may suggest clinical analyses that can consider the β3-AR blockers as novel treatments for melanoma and other tumors.

### 4.2. β3-AR in Breast Cancer

Regardless of the advances in cancer treatment over the past few years, breast cancer (BC) remains, still, a deadly disease. Since 1985, it has been shown the correlation between β-ARs and the mammary gland [205,206,207]. The first studies of β-AR stimulation were performed in normal bovine mammary glands for their effects on milk production [208,209,210] and were also described in the normal mammary gland and in mammary tumors of experimental animals [207,211]. Stress, chronic depression and social support might affect cancer initiation and progression [212], and β-ARs could be involved in these processes [213,214]. 

The paradoxical nature of β-AR activity in BC cells has recently been studied [215]. β-ARs were over-expressed in BC in relation to non-diseased breast epithelium [216]. In BC cell lines, pre-clinical studies have correlated the beta-blockers treatment with decreased cell proliferation and migration [217,218,219]. In the human BC cell line MDA MB-231, β-ARs were highly expressed and their stimulation induced a strong reduction in DNA synthesis [14]. In another clinical study, it was strengthened the β-ARs’ involvement in cancer, since it has been observed a reduction in metastatic dissemination and a BC-specific mortality in hypertensive women treated with propranolol [220]. Other researchers have shown that β-AR antagonists decreased BC progression, tumor metastasis, and patient mortality, but the underlying mechanisms are still under debate [215]. Moreover, the stimulation of β2-ARs induced significant tumor regression and suppression of tumor growth [221]. In an in vitro model of BC, propranolol increased the anti-angiogenic and anti-tumor efficacy of chemotherapic agents [222]. Moreover, some single nucleotide polymorphisms of *ADRB2* have been linked with higher BC risks [223]. Montoya et al. have shown that the three β-AR subtypes were expressed in BC tissue, with an over-expression of β1-AR and β3-ARs compared to normal breast tissue. In addition, in patients with early-stage BC, the use of non-selective β-blockers may lead to decreased tumor proliferation [216].

Interestingly, in BC, it has been reported the presence of both β3-AR mRNA and protein, and it has been studied the correlation with the β3-AR polymorphism Trp64Arg that has shown an increased susceptibility to endometrial cancer and a decreased risk of BC, specifically when associated with the Gln27Glu polymorphism in the β2-AR gene [224,225]. A correlation between the β2-AR-aberrant expression and the stimulation of the oncogenic properties of BC has been proposed, since it has been observed an increasing axillary lymph node metastasis leading to poor disease-free survival [226,227]. β2-ARs mediated these processes through the activation of the cAMP-calcium feed-forward loop that drives cell invasion [228,229]. While it is known that β2-ARs play a crucial role in BC, Montoya et al. showed that β1- and β3-ARs proteins in BC tissue are expressed more than in the normal mammary epithelium [216]. β1-and β3-ARs were reported in cancer processes, where β1-ARs promoted increased lipolysis in cancer cachexia [230] and β3-AR missense mutations correlated with obesity in BC in African Americans [231]. 

In another study, Zhou et al. showed that β3-AR was important in cell mobilization and differentiation. β3-AR expression was higher in BC tissues than in nearby non-cancerous tissues [232], and SR59230A inhibited the BC xenograft tumor growth. 

### 4.3. β3-AR in Retinopathy

At the retinal level, β-ARs cover an important role in angiogenesis regulation driven by hypoxia [233]. In specific, β3-ARs control various vascular functions and regulate pathological angiogenesis in models of retinal vascular proliferation. The oxygen-dependent angiogenesis is particularly important in this tissue; indeed, the retina is characterized by a high amount of oxygen consumption. Angiogenesis under hypoxia is essential for retinal vascularization during development, but it is also involved in the pathogenesis of numerous retinal diseases [234].

In the human retina, the expression of the β3-AR protein was primally detected in cultured retinal endothelial and choroidal cells where the stimulation by the agonist promoted cellular growth, migration, invasion and elongation [235]. In retinal diseases of retinopathy of prematurity (ROP) were observed high levels of β3-ARs expression [236]. In the hypoxic retina, β3-AR was localized in the proliferating vessels, where it was strongly upregulated under low oxygen tension. It has been shown that hypoxia induced an increase inf NA production and β3-AR upregulation in the endothelial cells of the retina, whose activation promoted invasion, migration and proliferation of human retinal cells [237]. This suggested that β3-AR antagonism could induce vessels’ proliferation in retinopathies through NO production and cyclic GMP (cGMP) accumulation, which increased VEGF release [238].

The selective β3-AR inhibition had a pro-proliferative effect through the mitogen-activated protein kinase (MAPK) pathway, while cell migration induced by β3-AR relied on an Src-PI3K pathway. Moreover, β3-AR has been correlated with de novo angiogenesis in retinal vascular proliferation [239].

At present, in rodent models, studies are focusing on the possible role of β3-ARs as a therapeutic target in ROP [236], where they revealed to have an important and original contribution with proangiogenic effect [172,233,240].

In β1/β2-ARs-KO mice models that were resistant to oxygen-induced retinopathy, after β3-ARs activation, the retinal angiogenesis was induced more than wild-type mice. This implied that β3-ARs are usually under-activated but, if adequately stimulated, they turn into active and able to replace β2-ARs in sustaining the angiogenic drive [240]. Moreover, it has been observed that β3-AR in rodent retinal blood vessels exerts vasorelaxant and antiangiogenic effects [241]. 

Hypoxia is significantly involved in the regulation of retinal vascularization-associated diseases by triggering neo-angiogenesis through VEGF release; thus, in mouse retinas, several studies focused on the production of VEGF through β3-AR activity. Indeed, Dal Monte et al. showed the increased levels of β3-AR protein in hypoxic retinas and that the siRNA silencing of the receptor reduced the increase of VEGF under hypoxia [240]. The stimulation of β3-ARs increased the VEGF release in hypoxic retinas compared to normoxic untreated ones, but their blockade prevented VEGF increase, suggesting a potential role for β-blockers in the retinal disease treatment. In addition, findings that hyperoxia downregulated retinal β3-ARs, in combination with HIF-1α and VEGF reduction, suggested the probable involvement of β3-ARs in response to high oxygen tension. 

Amato et al., in a mouse model of retinopathy, showed increased expression of β3-ARs through the HIF-1 pathway that binds to the enhancer region of the receptor [186]. They observed that β3-AR expression, similarly to VEGF, depended on oxygen tension. The finding that β3-ARs were upregulated in the retina in response to hypoxia, in association with HIF-1α, was in line with previous findings showing the possibility that increased levels of β3-AR would be coupled to the retinal angiogenic response [236,242].

Even though the role of β3-AR in the angiogenesis of retina remains, so far, to be clarified, these preliminary findings implied the influence of β3-AR on massive vessel proliferation in response to hypoxia.

### 4.4. β3-AR in Prostate Cancer

The involvement of β-ARs signaling has been shown even in mouse models of prostate carcinomas [243,244], where the β-ARs antagonism blocked the improvement of tumor progression/metastasis induced by stress. Conversely, still in prostate cancer, in the absence of stress in vivo, β-ARs agonism accelerated tumor progression and metastases [214,245]. Behavioral stress increased the β-AR signaling, which promoted tumor angiogenesis by inducing the epigenetic regulator histone deacetylase-2 (HDAC2) [246,247]. The group showed that HDAC2 was the exact target of the cAMP response element binding protein (CREB), which was activated by β-ARs signaling, for angiogenesis, in a xenograft model of prostate cancer in mice.

In prostate cancer, the β-AR signaling already linked to metastasis via inflammatory and circulatory mechanisms could provide the high detection of differences among groups with low progression/recurrence rates, which is a characteristic of early stage disease [3]. 

In guinea pigs, the isoprenaline reduced the contractions of prostate smooth muscles caused by electrical-field stimulation (EFS), which were antagonized by β1-ARs antagonists [248], whereas, in rats, both the β1 and β2-ARs caused the prostate relaxations [249]. In pre-clinical models of prostate cancer, it has been found that the sympathetic nervous system effects were mediated principally by β2- and β3-ARs [3,214,230]. In another study by Keng et al., it has been studied the β3-ARs’ agonism as a new treatment in men with benign prostatic hyperplasia (BPH) [250]. The group sustained that the therapy with β3-ARs agonists did not have additional effects on the urological symptoms if compared to present standard BPH treatment and, despite the β3-AR agonists’ improved the urological symptoms, it must have been for the difference in pharmacologic profile. At the same time, the β3-AR agonists did not increase the urinary retention rate or adverse events rates. In cultured prostatic stroma cells, β1 and β3-ARs agonists, xamoterol and BRL37344, respectively, inhibited the increase of [Ca2þ]i, which was reverted by the β2 and β3-ARs antagonists [251]. Calmasini et al., via immunohistochemical analysis, evidenced the β3-ARs’ presence in the smooth muscle layer in the human prostate transition zone that surrounds the prostatic urethra and contains the mucosal glands. Moreover, they evaluated the effects of mirabegron in the prostate, isolated from either healthy rabbits or BPH patients [130], demonstrating that the β3-ARs agonist clearly counteracted the contractions of prostate smooth muscles induced by EFS and α1-adrenoceptor activation. The mirabegron efficacy was clearly reduced by the β3-ARs antagonist L748,334, but not by the β1 and β2-ARs antagonists, evidencing that, in prostate smooth muscle, mirabegron acted through β3-AR activity.

### 4.5. β3-AR in Neuroblastoma

The involvement of β-ARs has been shown in NB tumor, which is the most common extracranial solid tumor in infants. A typical characteristic of NB is the neuroendocrine capacity to secrete CA, which via β-AR may influence different signaling pathways in the TME [252]. NB patients present high levels of CA that reflect the severity of the malignancy [253]. Dopamine and NA can promote, in NB cells, apoptosis through GPCR-mediated signaling [254]. The available public database data have shown that *ADRB2* expression in primary NB tumors is correlated with prognosis or known biological risk factors [115,255]. In NB mouse models, both non-selective and selective β1/β2-AR blockers increased the chemotherapy response and exerted antitumor activity [256,257]. 

β3-ARs expressed on several NB cell lines and biopsies from patients have been shown to be involved in the tumor growth of NB in a syngeneic murine model [258]. In this study, Bruno et al. clearly showed that β3-AR expression and modulation strongly affected tumor growth via the SK2/S1P2 axis responsible for β3-AR-dependent effects. In another study, in clinical NB tissues, Deng et al. demonstrated the significant increase in β3-ARs compared with less malignant tumors, such as malignant ganglioneuroma and ganglio-neuroblastoma tissues, suggesting that the β3-ARs pathway was deregulated and overactivated [259]. Moreover, the β3-AR blockade inhibited NB cell proliferation and colony growth, suggesting that the inhibition of the β3-AR pathway suppressed NB cell growth in vitro. In the same study, β3-AR expression was analyzed in different clinical stages of NB, showing that it was more upregulated in high-grade clinical stages, showing that the β3-AR pathway may control NB cell metastasis.

Zheng et al. showed that β3-AR expression was significantly increased in NB tissues and that β3-AR-pathway blockade inhibited cell growth via mTOR signaling suppression that enlightened a novel regulatory axis of β3-AR and mTOR in NB cells [260]. Moreover, data have shown the pro-tumoral role of β3-AR in NB and its involvement in tumor growth in a syngeneic murine model [184]. In the murine Neuro2A cell line, both the antagonism of β3-AR and β3-AR silencing inhibited cell growth and progression, in particular, the antagonism reduced the stemness markers’ expression but enhanced the markers of differentiation [258]. In fact, the reduction of the CA levels in the plasma derived from NB mice after β3-AR blockade or SK2 inhibition strengthened the results of the S1P/β3-AR crosstalk in NB. 

Bruno et al., in a murine syngeneic model of NB, have studied if the β3-AR modulation could influence the response of the host immune system against the tumor [261]. In specific, to investigate the β3-ARs’ involvement in PD-L1 expression and in the immune reactivity in the TME. In A/J mice, the syngeneic Neuro2A NB cells were inoculated with the β3-AR antagonist, a PD-L1 monoclonal antibody (αPD-L1), or their combination. This study showed that β3-AR blockade led to the reactivation of the immune response, partly dependent on the involvement of the PD-1/PD-L1 signaling axis. Indeed, β3-AR antagonism on tumor-infiltrating lymphocytes (TILs) reduced their ability to produce IFN-γ, which reduced the expression of PD-L1 provoked by TILs infiltration into NB tumor cells. Furthermore, genomic analysis of NB patients showed that high *ADRB3* expression, compared to the low expression group, was correlated with worse clinical outcomes and that *ADRB3* expression affected different immune-related pathways [261]. In conclusion, in the TME of NB, β3-ARs could control the tumor and host immune system interaction, where its antagonism affected multiple pro-tumoral signaling pathways. These data accorded with other preclinical results, where a TME supplemented with NK, DC, and CD8+ T cells after the β3-AR blockade or αPD-L1 administration was functionally able to control NB tumor growth [262,263]. Before, it has been studied the β3-ARs’ role in the immune response of melanoma, where both SR59230A and propranolol neutralized melanoma growth in vivo and their effect were associated with a considerable increase in NK and CD8 cells, a strong reduction in Treg cells and MDSC within the tumor mass [156]. Another previous study by Calvani et al. has demonstrated the involvement of β3-ARs in fetal and tumor immune tolerance [264], where the TME reactivated fetal abilities, including immunosuppression, mostly through β3-AR activation.

However, until now, it has not been completely understood if β3-AR is expressed in NB and its role in NB tumor biology; thus, more studies are needed to better understand the mechanisms underlying the complex interaction linking the host immune system and the tumor counterpart in NB. 

### 4.6. β3-AR in Ewing Sarcoma

Ewing sarcoma (ES) is one of the most common pediatric malignant tumors, reported to represent about 2% of all childhood cancers [265]. In ES, the aggressive tumor cells have shown high levels of stress, the source of which is poorly understood. The fine regulation of reactive oxygen species (ROS) production is fundamental for ES tumor growth [266].

The first evidence of the β-AR-AC pathway in ES cells was observed in the 1980s [267,268]. As previously reported, stress is significantly involved in tumorigenesis, through CAs acting at the β-AR level, including β1- β2- and β3-ARs, where the β-AR antagonists seem to reduce tumor growth and progression. The presence of β3-AR protein has been observed in several pediatric tumors, such as infantile hemangioma, neuroblastoma, Ewing sarcoma, and osteosarcoma [172,184,258,269]. As recently showed by Calvani et al., β3-ARs expressed in the mitochondria of melanoma cells could drive the UCP-2 activity and could drive the ROS amount in the mitochondria [198], showing thus the antioxidant activity of β3-ARs through the β3-AR/UCP-2 axis. In another study, it was showed the dual role of β3-ARs in response to antioxidants, where the adrenoceptor directly inhibited NADPH–oxidase activity and induced catalase expression [175]. In addition, NA could induce the catalytic subunit of the glutamate–cysteine ligase, increasing the intracellular levels of glutathione (GSH) through stimulation of β3-AR in U-251 MG cells. In this work, β3-ARs induced GSH synthesis to maintain GSH homeostasis in glioma cells [270]. All these data supported the evidence of the antioxidant properties of β3-ARs. Another report by Calvani et al. further strengthened the antioxidant role of β3-AR in the human ES A673 cell line, where the β3-ARs were identified as the principal regulators of the cellular response to oxidative stress (OS) in cells under treatment with different micronutrients [271]. In specific, in ES cells, β3-ARs worked as an ROS sensor that controlled cells by inducing (or not) the antioxidant response to cell death. Since β3-AR antagonism led to massive cell death, blocking β3-AR in A673 cells could significantly increase the ROS levels by the toxic limit, resulting in cell death. Moreover, SR59230A strongly reduced cancer cell vitality, showing that the β3-ARs antagonism could be a novel therapeutic strategy for ES treatment due to its ability to decrease the antioxidant activity.

Whilst on this theme, it is known that, in ES cells, the balance between OS and antioxidants plays an important role in the efficacy of cancer therapy, and since ES cells are recognized to be susceptible to increased OS levels, Pasha et al. have proposed β3-AR as a putative target for a potential strategy in cancer therapy [269]. In this work, the group evaluated, in vitro, the effects of a nutraceutical antioxidant, apigenin, in ES cells. The authors observed that apigenin inhibited the antioxidant protein expression, such as superoxide dismutase-2, catalase, and thioredoxin, but it increased the UCP2 and GSH levels that, inversely, were strongly inhibited by β3-ARs antagonism. The antioxidant role of β3-ARs through the UCP2 protein could control the redox homeostasis in ES cells. The break in UCP2 signaling, together with the β3-ARs activity inhibition, caused an excessive increase in ROS production within the cellular microenvironment, which led to excessive OS, resulting thus in substantial cell death. 

Furthermore, the β-AR expression in pediatric OS tumors and in neural macrophages has been evaluated, observing that only the β2- and β3-ARs were expressed in the neural macrophages of a syngeneic OS murine model [184]. 

In another study by Calvani et al. in ES, the preliminary results showed the higher β3-AR expression in CD99+ cells (an antigen and marker of CTCs isolation and detection in ES [272]) in peripheral blood and cells from patients with metastasis or relapsed ones, suggesting a potential role of β3-AR as a predictive maker of disease or recurrence in both patients [273]. In detail, the group observed that metastatic patients with low β3-ARs levels in circulating CD99+ cells had a complete remission in the two years of the study, while metastatic ones, with higher β3-ARs in CD99+ cells, had relapsed within the study, indicating thus a possible role of β3-ARs in the ES progression. Other studies, previously, have reported that metastatic status is correlated with CTC levels and high β3-ARs in the CTCs of ES patients [274].

Based on these initial data, the group assumed that β3-ARs should be investigated as a potential predictive marker of relapse disease. Since β3-AR expression has been observed also on microenvironmental cells’ surface, the investigation of the β3-ARs levels in the microenvironment could be considered a valuable marker of “stromal malignancy”, in the meaning of an indicator of the active role of stromal cells in disease progression. In conclusion, since β3-ARs have been shown to be highly expressed in adipose tissue and since the TME has a key role in cancer progression [275], the group speculated that adipose tissue, in a TME with high β3-AR levels, could facilitate progression of ES tumors. 

Further studies need to be conducted in the future, as few in vivo or in vitro studies have correlated β3-ARs with ES.

## 5. Conclusions and Future Perspectives

Since its discovery, β3-AR has turned rapidly into an interesting target for novel therapeutic approaches. β3-ARs have gained great interest in research and clinical studies for their presence in several tissues, implying their important role in drug development. However, β3-ARs’ role in humans is somehow controversial due to the late discovery, the lack of selective detection instruments and the inter-species differences. Some of the β3-ARs’ characteristics, such as the limited tissue expression and its low inclination to desensitization, are elements that would facilitate sustained pharmacological stimulation with low systemic off-target effects. 

Together, all the scientific data reported in this review display the intriguing role of β3-ARs expressed by different type of pediatric and adult tumors with their pleiotropic effects. 

Recent discoveries by our group have updated the field. Our studies have showed the main roles of β3-ARs in different tumors, in particular pediatric ones [16,184,198,258,261,269,271]. β3-ARs in the TME could modulate the interaction between the tumor and host immune system in tumors where β3-AR blockade affected multiple pro-tumoral signaling pathways via removing the barrier induced by the cooperation of immune checkpoints and the tumor cell pro-survival pathways [261]. 

It should be underlined that the immune system modulation through the β-AR signaling supported the normal immune response in physiologic conditions; instead, in altered processes, the prolonged stimulation of SNS/CA/β-ARs may have damaging consequences where the constant stimulation of the adrenergic signaling could promote pathological processes, including cancer progression [276]. 

In another study of melanoma, our group supported the hypothesis that β3-ARs could promote immune tolerance [156]. The beneficial effects of β3-ARs blockade on the immune system and tumor outcome could represent a new approach to treat cancer immune-editing, and it could be an efficient therapy for melanoma treatment. As showed in another work by us, in the TME, β3-ARs could control the tumor stroma differentiation, suggesting a new strategy to neutralize melanoma progression by increasing together differentiation processes and cell death [277]. 

Research advances over the last two decades have proposed β3-ARs targeting for the treatment of various conditions, with some drugs undergoing phase II and III clinical trials. However, in the light of what has been demonstrated, the study of the β-AR signaling in the TME of tumors should be taken into high consideration. 

Altogether, targeting β3-AR in the TME could represent a promising approach for the multiple non-redundant pro-tumoral signaling pathways. Further studies are required to explain the underlying mechanisms and interactions between the TME and host immune system.

In conclusion, this review highlights all the data that together strengthen the pharmacologic theory of β3-AR as a promising therapeutic target in different treatments, supporting that this is the right path for the treatment of several conditions. 

## Figures and Tables

**Figure 1 biomolecules-14-00159-f001:**
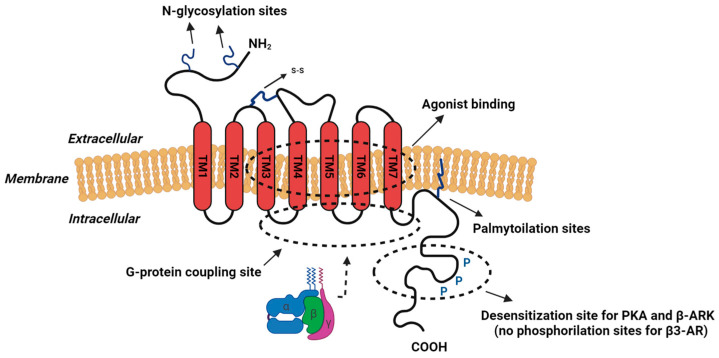
Structure of β-ARs.

**Figure 2 biomolecules-14-00159-f002:**
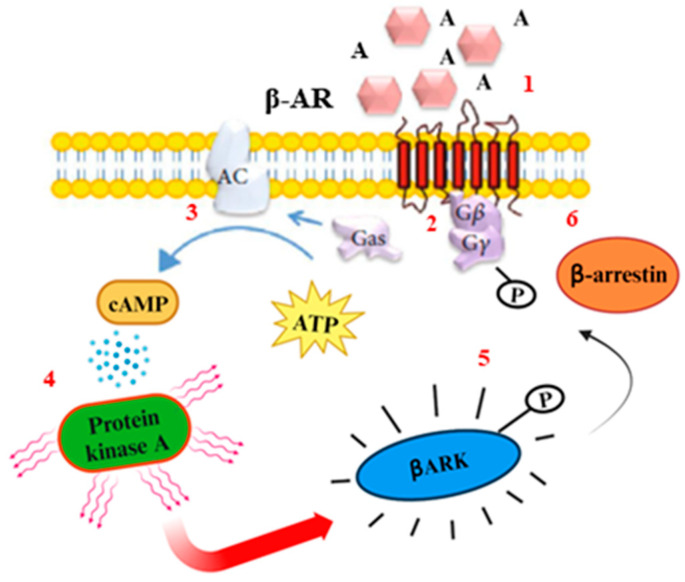
β-AR pathway. β-AR activation steps. 1: β-AR stimulation and activation of the Gs protein. Steps 2 and 3: the α subunit of the Gs protein stimulates adenylate cyclase (AC), which produces cAMP. Steps 4: cAMP activates the protein kinase (PKA), which phosphorylates β-AR kinase (β-ARK). Step 5: β-ARK phosphorylates the β-ARs residues. Step 6: In conclusion, β-arrestin finds the binding sites to bind the receptor.

**Figure 3 biomolecules-14-00159-f003:**
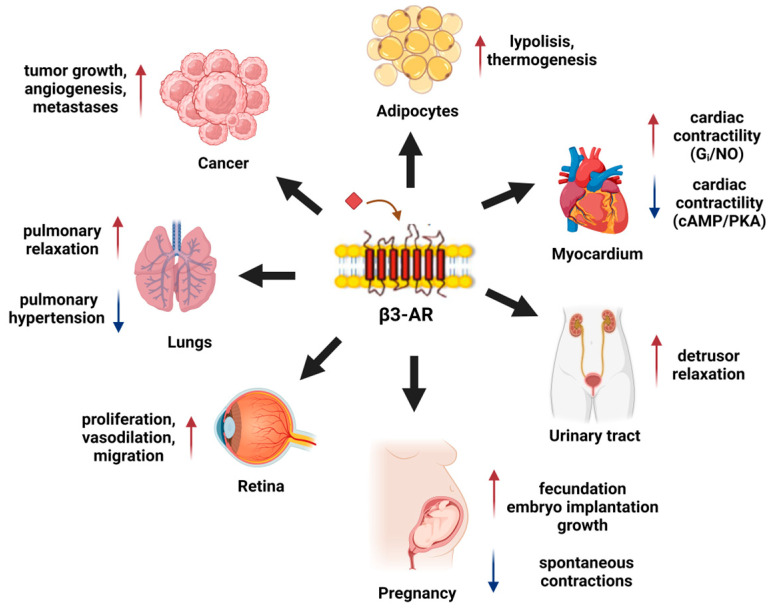
β3-ARs’ role in different tissues.

## Data Availability

Not applicable.

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
