# Peer review of "Inside the Biology of the β3-Adrenoceptor"

_biomolecules, 2024, doi:10.3390/biom14020159_

Round 1
Reviewer 1 Report
Comments and Suggestions for Authors
Reviewing the manuscript entitled « Inside the biology of β3-Adrenoceptor», authors performed a review on the biological activity of β3-AR. Since its discovery in 1989, β3-Adrenoceptor (β3-AR) has been a significant regulator of various physiological and metabolic functions, including thermogenesis, lipolysis, cardiomyocyte negative inotropic effects, blood vessel relaxation, and urinary bladder relaxation. It has also been proposed as a potential target for cancer treatment, as its up-regulation in tumor microenvironment under hypoxia regulates stromal cell differentiation, tumor growth, and metastases. Advances in research have led to the development of drugs targeting β3-AR for various conditions, with some currently in phase II and III clinical trials.
This paper is quite important and can be published after solving some minor issues.
· In section 2, where the types of receptors are described, I would suggest inserting a figure of their structures.
· There are several data in the literature supporting the involvement of the β3-AR receptor in prostate cancer (10.5213/inj.2142068.034, 10.3390/ijms21217958, 10.1158/1078-0432.CCR-11-0641). I would suggest inserting the corresponding paragraph in section 4.
Comments on the Quality of English Language
Minor editing of the English language is required.
Author Response
Dear reviewer, thank you for all your suggestions. Please see the attachment.
Kind regards
Maura Calvani

Reviewer 2 Report
Comments and Suggestions for Authors
The review “Inside the biology of β3-Adrenoceptor” by Pasha et al is exhaustive, a little too long and has redundancies (which should be avoided).
The current plan distinguishing physiology and pathology is artificial since, for example, pathological questions are addressed in the chapter dealing with β3-AR in adipose tissue and the myocardium. Another way of presenting the information could be to combine all the data (physiology, pathology and therapeutic use) organ by organ in a single paragraph.
While some parts of the manuscript read well in good English, others need to be revised extensively for the language. It appears that the manuscript results from the compilation of different parts written by different people. I'm fine with that, but in this case an author should take responsibility for revising the language and style of the entire manuscript.
Specific points
Many sentences should be rephrased for more clarity and some specific points should be corrected. Examples below
Line 66 “The β-AR is composed of 413 aminoacidic residues” replace with “the β2-AR comprises 413 aminoacid residues”
Line 71 “β-AR phosphorylation is involved in their rapid desensitization and then in their downregulation, allowing the control of receptor signalling”.
Line 77 “The downstream effects of activation of a GPCR depend on the type of Gα subunit (Gα.S, Gα.I, Gα.q and Gα.12), to which the receptor is coupled”
Line 86 “beta-arrestin binding to the receptor prevents further coupling to G proteins and promotes receptor internalization”
Line 100 “and they are also expressed in blood vessels and coronary arteries, where they promote increased irrigation of the organs”.
Line 106 “and differ from the β1- and β2 subtypes with respect to sequence and pharmacological profile”
Line 110 “the gene coding for ß3AR” or “the gene encoding ß3AR”
Line 114 “The ADRB3 gene, coding for the β3-AR, is made of a single intron which corresponds to a polypeptidic chain of 408 amino acid residues.
Line 119 “The C-terminus is the region with the lowest homology and is characterized by the absence of the phosphorylation sites, which are involved in the agonist-dependent desensitization of the β1 and β2 isoforms.”
Line 121 “The lack of phosphorylation sites thus makes β3-ARs resistant to agonist-induced desensitization” In this context, ref [38] is not appropriate and should be replaced by [36: Nantel et al. 1993]
Line 126 “It inhibits contractile activity…”
Line 132 “but their assignment to either one or the other category remains controversial” the meaning of this sentence is not clear
Line 136 “more recent” seems more appropriate than “late”
Line 232 ” In cardiac muscle β1 and β2-ARs are coupled to Gi”: To my knowledge, while all b-ARs are associated with Gas activation, b2-AR and b3-AR can also be coupled to inhibitory G protein according to studies which showed that their activity was pertussis toxin sensitive.
Comments on the Quality of English LanguageTo be revised as indicated
Author Response

(The authors gave the same response as above.)

Round 2
Reviewer 2 Report
Comments and Suggestions for Authors
Except for most of the specific points that have been addressed by the authors, the already identified drawbacks of the manuscript remain.
Since the article is a review article, if the authors decide to keep the current plan/structure of the manuscript, this will be under their responsibility
I will just focus on some factual issues that should be corrected before publication
Line 65: Again (already mentioned in my previous review), the 3 ß-adrenoceptor subtypes display different lengths; 413 residues is the length of the ß2AR. Writing “The β-AR comprises 413 aminoacid(ic) residues” is therefore not correct. If the authors choose a general sentence they should change the sentence. For example: ßARs are members of the 7-transmembrane…
Line 74: In general, it is assumed that the ligand binds to the receptor and not the other way around
Line 75: not necessarily the intracellular loop. In this case is the C-terminal tail
Line 80: The ligand stimulation of beta adrenoceptors (not the alpha)
Line 95: the ß2AR display a higher affinity for adrenaline (A) NOT for noradrenaline (NA)
Line 111: the authors likely mean “51% and 46% sequence homology with…
Comments on the Quality of English LanguageThe article should definitely be edited in deep for English and spelling errors
Author Response
Dear reviewer,
we appreciated your suggestions. Please see the attachment.
